

# LinkPred: a high performance library for link prediction in complex networks

Said Kerrache

Department of Computer Science, College of Computer and Information Sciences, King Saud University, Riyadh, Riyadh, Saudi Arabia

## ABSTRACT

The problem of determining the likelihood of the existence of a link between two nodes in a network is called link prediction. This is made possible thanks to the existence of a topological structure in most real-life networks. In other words, the topologies of networked systems such as the World Wide Web, the Internet, metabolic networks, and human society are far from random, which implies that partial observations of these networks can be used to infer information about undiscovered interactions. Significant research efforts have been invested into the development of link prediction algorithms, and some researchers have made the implementation of their methods available to the research community. These implementations, however, are often written in different languages and use different modalities of interaction with the user, which hinders their effective use. This paper introduces LinkPred, a high-performance parallel and distributed link prediction library that includes the implementation of the major link prediction algorithms available in the literature. The library can handle networks with up to millions of nodes and edges and offers a unified interface that facilitates the use and comparison of link prediction algorithms by researchers as well as practitioners.

## INTRODUCTION

The field of complex networks, and more generally that of network science, aims at studying networked systems, that is, systems composed of a large number of interacting components (*Albert & Barabási, 2002*). Under this umbrella fall many seemingly disparate networks, but which share common underlying topological properties that constitute a fertile ground for analyzing and ultimately understanding these systems. Networks of interest can be social, biological, informational, or technological. Link prediction is the task of identifying links missing from a network (*Lü & Zhou, 2011*; *Martínez, Berzal & Cubero, 2017*; *Guimerà & Sales-Pardo, 2009*; *Al Hasan et al., 2006*; *Guimerà & Sales-Pardo, 2009*; *Clauset, Moore & Newman, 2008*; *Lü & Zhou, 2011*; *Cannistraci, Alanis-Lobato & Ravasi, 2013*; *Daminelli et al., 2015*; *Al Hasan et al., 2006*; *Wang, Satuluri & Parthasarathy, 2007*; *Zhang et al., 2020*; *Beigi, Tang & Liu, 2020*; *Sajadmanesh et al., 2019*; *Makarov et al., 2019*), a problem with important applications, such as the reconstruction of networks from partial observations (*Guimerà & Sales-Pardo, 2009*), recommendation of items in online shops

Corresponding author
Said Kerrache, skerrache@ksu.edu.sa

and friends in social networks (*Al Hasan et al., 2006*), and the prediction of interactions in biological networks (*Clauset, Moore & Newman, 2008*).

This paper introduces LinkPred, a C++ high-performance link prediction library that includes the implementation of the major link prediction algorithms available in the literature by development from scratch and wrapping or translating existing implementations. The library is designed with three guiding principles, ease of use, extensibility, and efficiency. To facilitate its use, LinkPred borrows heavily from the STL design to offer an elegant and powerful interface. C++ users with minimum experience using STL will find the library's programming and usage style to be very familiar. Moreover, the use of templates allows for greater flexibility when using LinkPred and allows for integration within various contexts. The library contains bindings to Java and Python, providing access to its main functionalities through easy-to-use classes. LinkPred is aimed not only at practitioners but also at researchers in the field. It is designed to allow developers of new link prediction algorithms to easily integrate their code into the library and evaluate its performance. Efficiency-wise, the data structures used and implemented in LinkPred are all chosen and designed to achieve high performance. Additionally, most code in LinkPred is parallelized using OpenMp, which allows taking advantage of shared memory architectures. Furthermore, a significant portion of the implemented predictors supports distributed processing using MPI, allowing the library to handle very large networks with up to hundreds of thousands to millions of nodes.

In the rest of this paper, an overview of related software packages is presented first, followed by a description of the library's architecture and main functionalities. Example use cases with fully working code samples are presented next. The paper is then concluded by showing performance results and a comparison against existing link prediction packages.

## RELATED WORK

Several researchers in the area of link prediction have released implementations of their methods (*Clauset, Moore & Newman, 2008*; *Guimerà & Sales-Pardo, 2009*; *Liu et al., 2013*; *Papadopoulos, Psomas & Krioukov, 2015*; *Muscoloni & Cannistraci, 2017*). These implementations are, naturally but inconveniently, written in different languages and offer diverse modalities of interaction with the user, which complicates their effective use. There is also a limited number of packages that provide unified interfaces to implementations of topological ranking methods. The R package *linkprediction* (*Bojanowski & Chrol, 2019*), for instance, includes the implementation of the most important topological similarity algorithms. It offers a single method to compute the score of negative links using a specified similarity index. This package has several limitations, however. First, it only accepts connected undirected networks, which may be highly constraining as most real networks are disconnected. Since the package computes all negative links' scores, the size of networks that the package can handle is also limited. Furthermore, *linkprediction* does not offer any performance evaluation or test data generation functionalities. Other available packages include the commercial graph platform Neo4J (*Neo4J, 2019*), and NetworkX (*Hagberg, Schult & Swart, 2019*), which both contain the implementation of a limited number of

topological ranking methods. The Python package linkpred (*linkpred, 2020*) contains the implementation of a number of topological similarity methods and also global methods, including rooted PageRank, Katz index (*Katz, 1953*), and SimRank (*Jeh & Widom, 2002*). The library does not, however, support parallel and distributed implementations, nor does it support performance evaluation functionalities.

GEM (*Goyal & Ferrara, 2018b*; *Goyal & Ferrara, 2018a*) is a Python package that implements many state-of-the-art graph embedding techniques, including Locally Linear Embedding (*Roweis & Saul, 2000*), Laplacian Eigenmaps (*Belkin & Niyogi, 2001*), Graph Factorization (*Koren, Bell & Volinsky, 2009*; *Ahmed et al., 2013*), Higher-Order Proximity preserved Embedding (HOPE) (*Ou et al., 2016*), Structural Deep Network Embedding (SDNE) (*Wang, Cui & Zhu, 2016*), and node2vec (*Grover & Leskovec, 2016*). It also includes several similarity measures that can be used in combination with these embedding algorithms to predict links. GEM is, however, more focused on graph embedding techniques than link prediction and, as such, does not include other types of link prediction methods such as topological similarity and probabilistic methods. SNAP (Stanford Network Analysis Platform) (*Leskovec & Sosič, 2016*), which is a general-purpose network analysis library, also includes an implementation of node2vec and GraphWave (*Donnat et al., 2018*). Like GEM, SNAP is not dedicated to link prediction, and apart from its graph embedding algorithms, it includes only a limited number of topological similarity measures as part of its experimental components (snap-exp). Another general-purpose network analysis library is the Python package scikit-network (*Bonald et al., 2020*), which contains the implementation of a number of local methods and graph embedding algorithms.

Given the importance of link prediction and the wide range of existing and potential applications, the currently available software packages clearly lack functionality and performance. Arguably, this state-of-affairs limits the successful application of existing algorithms to real-life problems and the rigorous testing of newly proposed methods. LinkPred aims at filling this gap that separates existing research from efficient software implementations. Table 1 contains a comparison in terms of functionality between LinkPred and the main open-source packages used for link prediction. The architecture of LinkPred and the functionalities shown in Table 1 are discussed in detail in the next section.

# ARCHITECTURE AND FUNCTIONALITIES

LinkPred aims at filling the existing gap between research and efficient software implementations of link prediction algorithms. As shown in Fig. 1, it offers functionalities at various levels to help use, implement and test link prediction methods. In this section, a brief description of the functionalities available in LinkPred is given. More details can be found in the library user guide.

## Core components

At the core of LinkPred lie efficient data structures for storing and accessing network data. These include the classes `UNetwork` and `DNetwork` used to represent undirected and directed networks, respectively. These structures allow efficient access to nodes, edges, and

Kerrache (2021), *PeerJ Comput. Sci.*, DOI 10.7717/peerj-cs.521

**Table 1** Comparison of LinkPred against the most important free/open-source link prediction software packages.

| Functionality | LinkPred | NetworkX | *linkprediction* | GEM | SNAP | linkpred | scikit-network |
|---|---|---|---|---|---|---|---|
| Supported languages | C++, Python (a subset of the functionalities), Java (a subset of the functionalities) | Python | R | Python | C++, Python (a subset of the functionalities) | Python | Python |
| Topological similarity methods | Yes (with shared memory and distributed parallelism) | Yes (no parallelism) | Yes (no parallelism) | No | No (A limited number of algorithms is included as an experimental component) | Yes (no parallelism) | Yes (no parallelism) |
| Global link prediction methods | Yes (with shared memory parallelism and for some predictors also distributed parallelism) | No | No | No | No | Yes (Rooted PageRank, SimRank, Katz, shortest path) | No |
| Graph embedding algorithms | LLE, Laplacian Eigenmaps, Graph Factorization, DeepWalk, LINE, LargeVis, node2vec, and HMSM | No | No | LLE, Laplacian Eigenmaps, Graph Factorization, HOPE, SDNE, and node2vec | node2vec and GraphWave | No | Spectral, SVD, GSVD, PCA, Random Projection, Louvain, Hierarchical Louvain, Force Atlas, and Spring. |
| Classifiers | Yes (mainly via mlpack) | No | No | No | No | No | No |
| Similarity measures | Yes | No | No | Yes | Yes | No | No |
| Test data generation | Yes | No | No | No | No | No | No |
| Performance measures | Yes | No | No | No | No | No | No |

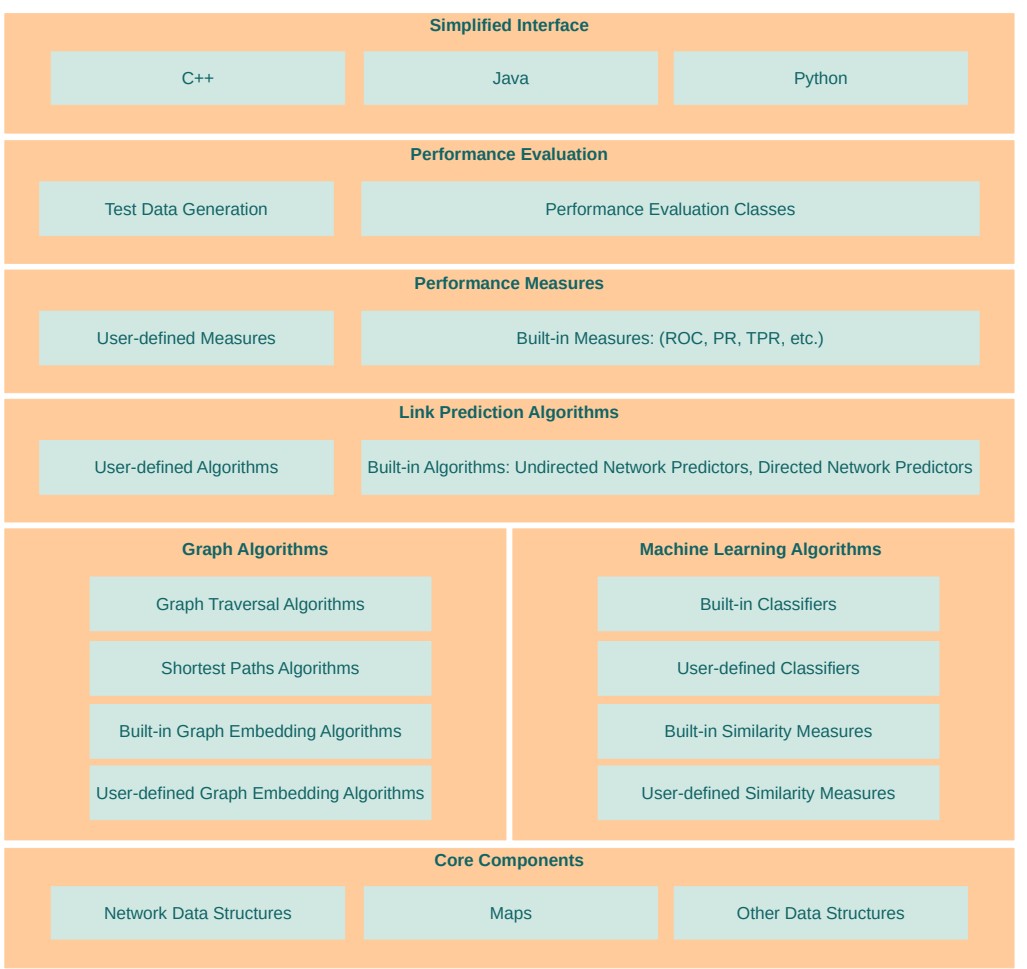

**Figure 1  Architecture of LinkPred.**

non-existing edges through C++-style iterators. Also included are auxiliary data structures such as full and sparse node and edge maps.

### The network data structures

The life cycle of a network has two distinct phases. In the pre-assembly phase, it is possible to add nodes and edges to the network. It is also possible to access nodes and translate external labels to internal IDs and vice versa. However, most functionalities related to accessing edges are not yet available. As a result, the network at this stage is practically unusable. To be able to use the network, it is necessary to assemble it first. Once assembled, no new nodes or edges can be added (or removed) to the network. The network is then fully functional and can be passed as an argument to any method that requires so.

To build a network, an empty network is first created by calling the default constructor:

```
UNetwork<> net;
```

Most classes in LinkPred manipulate networks through smart pointers for efficient memory management. To create a shared pointer to a UNetwork object:

```
auto net = std::make_shared<UNetwork<>>();
```

Notice that the class UNetwork is a class template, which is instantiated with the default template arguments. In this default setting, the labels are of type std::string, whereas internal IDs are of type **unsigned int**, but UNetwork can be instantiated with several other data types if wanted. For instance, the labels can be of type **unsigned int**, which may reduce storage size in some situations.

Adding nodes is achieved by calling the method addNode, which takes as parameter the node label and returns an std::pair containing, respectively, the node ID and a Boolean which is set to true if the node is newly inserted, false if the node already exists. The nodes IDs are guaranteed to be contiguous in $0, \ldots, n-1$, where $n$ is the number of nodes.

```
auto res = net.addNode(label);
auto id  = res.first; // This the node ID
bool inserted = res.second; // Was the node inserted or did it already exist?
```

The method addEdge is used to create an edge between two nodes specified by their IDs (**not their labels**):

```
net.addEdge(i, j);
```

The last step in building the network is to assemble it:

```
net.assemble();
```

The method assemble initializes the internal data structures and makes the network ready to be used.

Nodes can be accessed through iterators provided by nodesBegin() and nodesEnd(). For convenience, the iterator points to a pair, the first element of which is the internal ID, whereas the second is the external label.

```
std::cout << "ID\tLabel" << std::endl;
for (auto it = net.nodesBegin(); it != net.nodesEnd(); ++it) {
  std::cout << it->first << "\t" << it->second << std::endl;
}
```

Alternatively, one can iterate over labels in a similar way using the iterators labelsBegin() and labelsEnd():

```
std::cout << "Label\tID" << std::endl;
for (auto it = net.labelsBegin(); it != net.labelsEnd(); ++it) {
  std::cout << it->first << "\t" << it->second << std::endl;
}
```

It is also possible to translate labels to IDs and vice versa using getID(label) and getLabel(id) respectively. Oftentimes, one would want to iterate over a random sample of nodes instead of the whole set. This can be easily done using the two methods rndNodesBegin and rndNodesEnd.

Information on edges can only be accessed after assembling the network. One way to access edges is to iterate over all edges in the network. This can be done using the method edgesBegin() and edgesEnd(). As it is the case with nodes, it is possible to access a random sample of edges using rndEdgesBegin and rndEdgesEnd. LinkPred offers the possibility to iterate over negative links in the same way one iterates over positive edges. This can be done using the method nonEdgesBegin() and nonEdgesEnd():

```
std::cout << "Start\tEnd" << std::endl;
for (auto it = net.nonEdgesBegin(); it != net.nonEdgesEnd(); ++it) {
  std::cout << net.start(*it) << "\t" << net.end(*it) << std::endl;
}
```

It is also possible to iterate over a randomly selected sample of negative links using rndNonEdgesBegin and rndNonEdgesEnd.

To represent directed networks, LinkPred offers the class DNetwork, which offers a very similar interface to UNetwork.

### Maps

Maps are a useful way to associate data with nodes and edges. Two types of maps are available in LinkPred: *node maps* (class NodeMap) and *edge maps* (class EdgeMap), both member of UNetwork. The first assigns data to the network nodes, whereas the latter maps data to edges (see Fig. 2 for an example).

Creating a node map is achieved by calling the method createNodeMap on the network object. This is a template method with the mapped data type as the only template argument. For example, to create a node map with data type **double** over the network net:

```
auto nodeMap = net.template createNodeMap<double>();
```

Creating an edge map can be done in a similar way:

```
auto edgeMap = net.template createEdgeMap<double>();
```

Both NodeMap and EdgeMap offer the same interface, which in fact is similar to std::map. This includes the operator [], the methods at, begin, end, cbegin and cend. From the performance point of view, NodeMap offers constant time access to mapped values, whereas EdgeMap requires logarithmic time access ($O(\log m)$, $m$ being the number of edges).

If a node map is sparse, that is, has non-default values only on a small subset of the elements, it is better to use a sparse node map. To create a sparse node map:

```
auto nodeSMap = net.template createNodeSMap<double>(0.0);
```

Notice that the method takes as input one parameter that specifies the map's default value (in this case, it is 0.0). Hence, any node which is not explicitly assigned a value is assumed to have the default value 0.0.

## Graph algorithms

To facilitate the implementation of link prediction algorithms, LinkPred comes with a set of graph-algorithmic tools such as efficient implementations of graph traversal, shortest path algorithms, and graph embedding methods.

### Graph traversal and shortest paths algorithms

LinkPred provides two classes for graph traversal: BFS, for Breadth-First traversal, and DFS for Depth-First traversal. They both inherit from the abstract class GraphTraversal, which declares one virtual method traverse. It takes as parameter the source node from where the traversal starts and a reference to a NodeProcessor object, which is in charge of processing nodes sequentially as they are visited.

strategies to balance memory usage and computation, and `ASPDistCalculator`, an approximate shortest path distance calculator. The approximation used in `ASPDistCalculator` works as follows. A set $\mathcal{L}$ of nodes called *landmarks* is selected, and the distance from each landmark to all other nodes is pre-computed and stored in memory. The distance between any two nodes $i, j$ is then approximated by:

$$d_{ij} \simeq \min_{k \in \mathcal{L}}[d_{ik} + d_{kj}]. \tag{1}$$

The landmarks are passed to `ASPDistCalculator` object using the method `setLandmarks`. Naturally, by increasing the number of landmarks, more precision can be obtained, be it though at a higher computational and memory cost.

### Graph embedding algorithms

Graph embedding consists in transforming the graph's nodes and edges into elements of a low-dimensional vector space while preserving, as much as possible, its structural properties (*Goyal & Ferrara, 2018c*). It is a problem with important applications in various fields, including link prediction (*Goyal & Ferrara, 2018c*; *Kazemi & Poole, 2018*; *Alharbi, Benhidour & Kerrache, 2016*; *Wang, Cui & Zhu, 2016*), product recommendation (*Koren, Bell & Volinsky, 2009*), data visualization (*van der Maaten & Hinton, 2008*; *Tang et al., 2016*; *Cao, Lu & Xu, 2016*), and node classification (*Bhagat, Cormode & Muthukrishnan, 2011*; *Tang, Aggarwal & Liu, 2016*).

LinkPred contains several state-of-the-art graph embedding algorithms, some of which are implemented from scratch, whereas others are based on publicly available implementations. These include methods based on matrix decomposition, namely Locally Linear Embedding (*Roweis & Saul, 2000*) implemented in the class LLE, Laplacian Eigenmaps (*Belkin & Niyogi, 2001*) implemented in the class LEM, and Matrix Factorization (*Koren, Bell & Volinsky, 2009*) (also referred to as Graph Factorization in *Goyal & Ferrara, (2018c)*; *Ahmed et al., (2013)*) implemented in the class `MatFact`. Also available are methods based on random walks, including DeepWalk (*Perozzi, Al-Rfou & Skiena, 2014*) implemented in the class `DeepWalk`, Large Information Networks Embedding (LINE) (*Tang et al., 2015*), implemented in the class LINE, LargeVis (*Tang et al., 2016*) implemented in the class `LargeVis`, and node2vec (*Grover & Leskovec, 2016*), which is implemented in the class `Node2Vec`. Additionally, the librray includes the implementation of the Hidden the Metric Space Model (HMSM) embedding method (*Alharbi, Benhidour & Kerrache, 2016*) available through the class HMSM.

To provide a uniform interface, all embedding algorithms implemented in LinkPred inherit from the abstract class `Encoder`, which declares the following methods:

- The method `init`, which is first called to initialize the internal data structures of the encoder. This is a pure virtual method of the class `Encoder` and must be implemented by derived classes.
- Once the encoder is initialized, the method `encode`, also a pure virtual method, is called to perform the embedding. This step typically involves solving an optimization problem, which can be computationally intensive both in terms of memory and CPU

usage, especially for very large networks. The dimension of the embedding space can be queried and set using `getDim` and `setDim` respectively.

- The node embedding or the node code, which is the vector of coordinates assigned to the node, can be obtained by calling the method `getNodeCode`. The edge code is by default the concatenation of its two nodes' codes and can be obtained using `getEdgeCode`. Hence, in the default case, the edge code dimension is double that of a node. Classes that implement the `Encoder` interface may change this default behavior if desired. The user can query the dimension of the edge code using the method `getEdgeCodeDim`.

Having a unified interface for encoders allows embedding algorithms to be easily combined with different classifiers and similarity measures to obtain various link prediction methods, as explained in the next sections. It also allows users to use their own embedding algorithms to build and test new link prediction methods.

## Machine learning algorithms

The library contains the implementations of several classifiers and similarity measures that can be combined with graph embedding algorithms (see the previous section) to build a variety of link prediction methods. Available classifiers, most of which are derived from mlpack (*Curtin et al., 2013*), include logistic regression, feed-forward neural networks, linear support vector machine, and Naive Bayes classifier. All binary classifiers in LinkPred implement the interface `Classifier`, which provides two important methods: the method `learn` which trains the classifier on a training set, and the method `predict` which predicts the output for a given input.

Similar to classifiers, all similarity measures in LinkPred inherit from the abstract class `SimMeasure`, which defines one method, `sim`, which computes the similarity between two input vectors. Implemented similarity measures include cosine similarity, dot product similarity, $L_1$, $L_2$ and $L_p$ similarity, and Pearson similarity.

## Link predictors

LinkPred includes a large selection of link prediction algorithms which can be broadly classified into three categories: topological similarity methods, global methods, and graph-embedding techniques. In terms of topological similarity predictors, the library contains the implementations of the most known algorithms existing in the literature, including Common Neighbors, Adamic-Adard, Resource Allocation, Cannistraci Resource Allocation, and Jackard Index, among other predictors. (*Liben-Nowell & Kleinberg, 2007*; *Newman, 2001*; *Jaccard, 1901*; *Adamic & Adar, 2003*; *Ravasz et al., 2002*; *Papadimitriou, Symeonidis & Manolopoulos, 2012*; *Liu & Lü, 2010*; *Lichtenwalter, Lussier & Chawla, 2010*; *Yang, Yang & Zhang (2015)*; *Yang, Lichtenwalter & Chawla, 2015*; *Zhu & Xia, 2015*; *Muscoloni & Cannistraci, 2017*; *Cannistraci, Alanis-Lobato & Ravasi, 2013*; *Daminelli et al., 2015*). Due to their local nature, these algorithms can scale to very large networks, especially when executed on distributed architectures. Addiitonally, the library includes several state-of-the-art global link predictors, such as SBM (*Guimerà & Sales-Pardo, 2009*), HRG (*Clauset, Moore & Newman, 2008*), FBM (*Liu et al., 2013*), HyperMap (*Papadopoulos*

*et al., 2012*; *Papadopoulos, Psomas & Krioukov, 2015*) and the popularity-similarity method proposed in *Kerrache, Alharbi & Benhidour (2020)*.

LinkPred also supports link prediction algorithms based on graph embedding, where the network is first embedded into a low dimensional vector space, whereby nodes are assigned coordinates in that space while preserving the network's structural properties. These coordinates can be used either to compute the similarity between nodes or as features to train a classifier to discriminate between existing edges (the positive class) and non-existing edges (the negative class) (*Goyal & Ferrara, 2018c*). LinkPred provides two classes that can be used to build link prediction algorithms based on graph embedding: the class `UECLPredictor`, which combines an encoder (a graph embedding algorithm) and a classifier, and the class `UESMPredictor`, which pairs the encoder with a similarity measure as illustrated in Fig. 3.

In addition to algorithms for undirected networks, several adaptations of topological similarity methods to directed networks are available as well. The library offers a unified interface for all link prediction algorithms, simplifying the use and comparison of different prediction methods. The interface is called `ULPredictor` for predictors in undirected networks and `DLPredictor` for those in directed networks. Most implemented predictors support shared-memory parallelism, and a large number of them support distributed memory parallelism, allowing LinkPred to take advantage of the power of HPC clusters to handle very large networks.

### The predictor interface

As stated above, all link predictors for undirected networks must inherit from the abstract class `ULPredictor`. It declares three important pure virtual methods that the derivative classes must implement:

- The method **void** `init()`: This method is used to initialize the predictor's state, including any internal data structures.
- The method **void** `learn()`: In algorithms that require learning, it is in this method that the model is built. The learning is separated from prediction because, typically, the model is independent of the set of edges to be predicted.
- The method **double** `score(Edge **const** & e)`: returns the score of the edge e (usually a non-existing edge).

In addition to these three basic methods, `ULPredictor` declares the following three virtual methods, which by default use the method `score` to assign scores to edges, but which can be redefined by derived classes to achieve better performance:

- The method **void** `predict(EdgeRndIt begin, EdgeRndIt end, ScoreRndIt scores)`: In this method, the edges to be predicted are passed to the predictor in the form of a range (`begin`, `end`) in addition to a third parameter (`scores`) to which the scores are written. This is a virtual method that uses the method `score` to assign scores to edges and can be redefined by derived classes to provide better performance.
- The method `std::pair<NonEdgeIt, NonEdgeIt> predictNeg(ScoreRndIt scores)` predicts the score for all negative (non-existing) links in the network. The

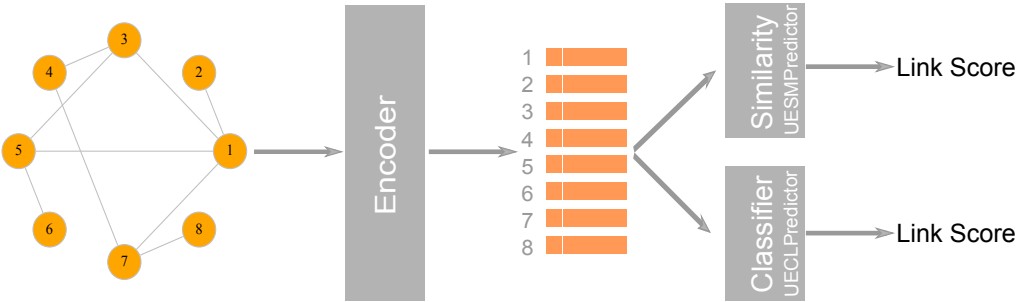

**Figure 3** **The first stage in a graph embedding method is accomplished by an encoder class which uses a graph embedding algorithm to assign coordinates to nodes.** In the class `UESMPredictor`, this is followed by a similarity measure to predict link scores, whereas `UESMPredictor` uses a classifier to make the prediction.

scores are written into the random output iterator `scores`. The method returns a pair of iterators begin and end to the range of non-existing links predicted by the method.

- The method `std::size_t top(std::size_t k, EdgeRndOutIt eit, ScoreRndIt sit)` finds the *k* negative edges with the top scores. The edges are written to the output iterator `eit`, whereas the scores are written to `sit`.

The class `ULPredictor` offers default implementations for the methods `top`, `predict` and `predictNeg`. Sub-classes may use these implementations or redefine them to achieve better performance.

The abstract class `DLPredictor` plays the same role as `ULPredictor` but for link predictors in directed networks. It offers the same interface as the latter but with different default template arguments and methods implementation.

## Performance evaluation

LinkPred offers a set of tools that help to streamline the performance evaluation procedure. This includes data setup functionalities, which can be used to create test data by removing and adding edges to ground truth networks. The library also includes efficient implementations of the most important performance measures used in link prediction literature, including the area under the receiver operating characteristic (ROC) curve, the area under the precision–recall (PR) curve, and top precision. The area under the PR curve can be computed using two integration methods: the trapezoidal rule, which uses a linear interpolation between the PR points, and the more accurate nonlinear interpolation method proposed in *Davis & Goadrich (2006)*. In addition to performance measures implementations, LinkPred contains helper classes, namely `PerfEvaluator` and `PerfEvalExp`, that facilitate the comparative evaluation of multiple link prediction algorithms using multiple performance measures.

All performance measures inherit from the abstract class `PerfMeasure`. The most important method in this class is `eval` which evaluates the value of the performance measure. The performance measure results are written to an object of type `PerfResults` passed as a parameter of the method. The class

PerfResults is defined as `std::map<std::string, double>`, which allows the possibility of associating several result values with a single performance measure.

An important class of performance measures is performance curves such as ROC and PR curves. They are represented by the abstract class `PerfCurve`, which inherits from the class `PerfMeasure`. The class `PerfCurve` defines a new virtual method `getCurve`, which returns the performance curve in the form of an `std::vector` of points. In the remainder of this section, more details of the performance measures implemented in LinkPred are presented.

### Receiver operating characteristic curve (ROC)

One of the most important performance measure used in the field of link prediction is the receiver operating (ROC) curve, in which the true positive rate (recall) is plotted against the false positive rate. The ROC curve can be computed using the class ROC. Figure 4A shows an example ROC curve obtained using this class.

The default behavior of the `ROC` performance measure is to compute the positive and negative edge scores and then compute the area under the curve, which may lead to memory issues with large networks. To compute the area under the curve without storing both types of scores, the class ROC offers a method that *streams* scores without storing them. To enable this method, call `setStrmEnabled(bool)` on the ROC object. To specify which scores to stream use the method `setStrmNeg(bool)`. By default, the negative scores are streamed, while the positive scores are stored. Passing `false` to `setStrmNeg` switches this. In addition to consuming little memory, the streaming method supports distributed processing (in addition to shared memory parallelism), making it suitable for large networks.

### Precision–recall curve

The precision–recall (PR) curve is also a widely used measure of link prediction algorithms' performance. In this curve, the precision is plotted as a function of the recall. The PR curve can be computed using the class PR. The area under the PR curve can be computed using two integration methods:

- The trapezoidal rule which assumes a linear interpolation between the PR points.
- Nonlinear interpolation as proposed by Jesse Davis and Mark Goadrich (*Davis & Goadrich, 2006*).

The second method is more accurate, as linear integration tends to overestimate the area under the curve (*Davis & Goadrich, 2006*). Furthermore, the implementation of Davis-Goadrich nonlinear interpolation in LinkPred ensures little to no additional cost compared to the trapezoidal method. Figure 4B shows an example PR curve obtained using the class PR.

### General performance curves

LinkPred offers the possibility of calculating general performance curves using the class GCurve. A performance curve is, in general, defined by giving the *x* and *y* coordinates functions. These are passed as parameters, in the form of lambdas, to the constructor of the

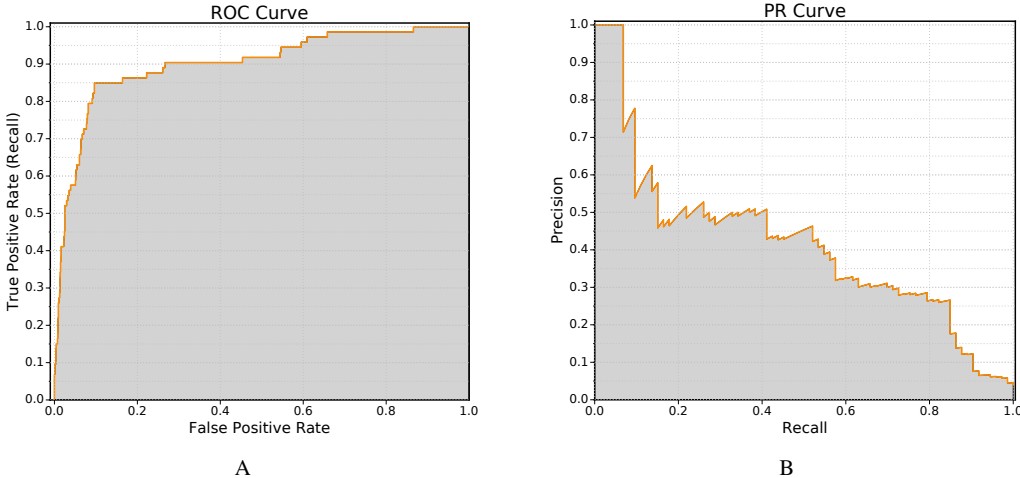

**Figure 4  Example of performance curves generated by LinkPred (the plots are created using an external tool).** The area under the curve (shown in gray) is the value associated with the performance curve.

class `GCurve`. The associated performance value is the area under the curve computed using the trapezoidal rule (linear interpolation). For example, the ROC curve can be defined as:

```
GCurve<> cur(fpr, rec, "ROC");
```

The two first parameters of the constructors are lambdas having the signature:

```
double(std::size_t tp, std::size_t fn, std::size_t tn, std::size_t fp, std::size_t
    P, std::size_t N)
```

### Top precision
The top precision measure is defined as the ratio of true positives within the top $l$ scored edges, $l > 0$ being a parameter of the measure (usually $l$ is set to the number of links removed from the network). Top precision is implemented by the class TPR, and since it is not a curve measure, this class inherits directly from `PerfMeasure`. The class TPR offers two approaches for computing top-precision. The first approach requires computing the score of all negative links, whereas the second approach calls the method `top` of the predictor. The first approach is, in general, more precise but may require more memory and time. Consequently, the second approach is the performance measure of choice for very large networks.

## Simplified interface and bindings
The simplified interface provides the essential functionalities available in LinkPred via a small number of easy-to-use classes. These classes are very intuitive and can be used with a minimum learning effort. They are ideal for initial use of the library and exploring its main functionalities. Java and Python bindings for the simplified interface are also available, facilitating the library's use by users who are more comfortable using these languages. The simplified interface contains two main classes: `Predictor`, which allows computing the scores for an input network using all available link prediction algorithms, and the

class `Evaluator`, which can be used for performance evaluation. Also included are simple structures to store prediction and performance results. These classes are designed in a simple way that allows uniform usage across different programming languages.

# EXAMPLE USE CASES

This section describes four main use scenarios of the library. The first use case demonstrates the working of the simplified interface in different languages, which is typical for first-time use of the library or for users who prefer to use the library in Python or Java. The second scenario consists in computing the scores of all non-existing links in a network, which is the typical use case for a practitioner working on networked data. Researchers in link prediction are typically interested in implementing new link prediction algorithms, which is presented as the third use case, and evaluating their performance, which is use case number four.

## Using the simplified interface

The first example program shows how to use the simplified interface to obtain the top *k* ranked edges using Adamic Adar index in C++:

```cpp
#include <linkpred.hpp>
#include <iostream>
using namespace LinkPred::Simp;
int main() {
  int k = 10;
  // Create a prtedictor object
  Predictor p;
  // Load network from file
  p.loadnet("Zakarays_Karate_Club.edges");
  // Predict the top k edges using Adamic Adar index
  std::vector<EdgeScore> esv = p.predTopADA(k);
  // Print the scores
  for (auto it = esv.begin(); it != esv.end(); ++it) {
    std::cout << it->i << "\t" << it->j << "\t" << it->score << std::endl;
  }
  return 0;
}
```

in Python:

```python
# Import the module
import LinkPredPython as lpp
k = 10;
# Create a predictor object
p = lpp.Predictor();
# Load network from file
p.loadnet("Zakarays_Karate_Club.edges");
# Predict the top k edges using Adamic Adar index
esv = p.predTopADA(k);
# Print the scores
for es in esv:
  print(es.i + "\t" + es.j + "\t" + "{:.4f}".format(es.score));
```

and finally, in Java;

```java
public class PredictorExp {
  static {
    // Load the library
    System.loadLibrary("LinkPredJava");
```

```
  }
  public static void main(String[] args) {
    int k = 10;
    // Create a prtedictor object
    Predictor p = new Predictor();
    // Load network from file
    p.loadnet("Zakarays_Karate_Club.edges");
    // Predict the top k edges using Adamic Adar index
    EdgeScoreVec esv = p.predTopADA(k);
    // Print the scores
    for (int i = 0; i < esv.size(); i++) {
      EdgeScore es = esv.get(i);
      System.out.println(es.getI() + "\t" + es.getJ() + "\t" + es.getScore());
    }
  }
}
```

The output of these three programs is as follows:

```
1   33  1.61374
1   34  2.71102
2   34  2.25292
3   32  1.67334
3   34  4.71938
5   6   1.99226
7   11  1.99226
8   14  1.8082
32  24  1.66562
24  25  1.63159
```

In the second example, the performance measure of several link prediction algorithms is evaluated by removing 10% of the links from the network and using it as a test set. The performance is assessed using two performance measures, area under the ROC curve, and top-precision. The code for this example in C++:

```
#include <linkpred.hpp>
#include <iostream>
using namespace LinkPred::Simp;
int main() {
  int nbRuns = 10;
  double edgeRemRatio = 0.1;
  // Create an evaluator object
  Evaluator eval;
  // Add predictors to be evaluated
  eval.addCNE();
  eval.addADA();
  eval.addKAB();
  // Add performance measures
  eval.addROC();
  eval.addTPR();
  // Run experiment on the specified network
  eval.run("Zakarays_Karate_Club.edges", nbRuns, edgeRemRatio);
  return 0;
}
```

in Python:

```
# Import the module
import LinkPredPython as lpp
nbRuns = 10;
edgeRemRatio = 0.1;
# Create an evaluator object
ev = lpp.Evaluator();
```

```
# Add predictors to be evaluated
ev.addCNE();
ev.addADA();
ev.addKAB();
# Add performance measures
ev.addROC();
ev.addTPR();
# Run experiment on the specified network
ev.run("Zakarays_Karate_Club.edges", nbRuns, edgeRemRatio);
```

and in Java:

```java
public class EvaluatorExp {
  static {
    // Load the library
    System.loadLibrary("LinkPredJava");
  }
  public static void main(String[] args) {
    int nbRuns = 10;
    double edgeRemRatio = 0.1;
    // Create an evaluator object
    Evaluator eval = new Evaluator();
    // Add predictors to be evaluated
    eval.addCNE();
    eval.addADA();
    eval.addKAB();
    // Add performance measures
    eval.addROC();
    eval.addTPR();
    // Run experiment on the specified network
    eval.run("Zakarays_Karate_Club.edges", nbRuns, edgeRemRatio);
  }
}
```

The output of these three programs is as follows:

```
#ratio  ROCADA  ROCCNE  ROCKAB  TPRADA  TPRCNE  TPRKAB
0.10    0.7737  0.7149  0.8280  0.1250  0.1932  0.1250
0.10    0.6593  0.6333  0.7030  0.1250  0.0000  0.1250
0.10    0.5967  0.5762  0.6095  0.1875  0.1818  0.2500
0.10    0.8464  0.7913  0.9343  0.1875  0.1290  0.3750
0.10    0.8324  0.7785  0.8967  0.1250  0.1750  0.1250
0.10    0.7240  0.6953  0.7547  0.0000  0.2222  0.0000
0.10    0.6753  0.6610  0.7262  0.0000  0.1591  0.1250
0.10    0.6048  0.5792  0.6672  0.0000  0.0000  0.0000
0.10    0.7627  0.7547  0.7808  0.2917  0.3194  0.3750
0.10    0.6442  0.5835  0.6727  0.1250  0.1250  0.1250
```

## Predicting missing links

When dealing with networked data, a data scientist may be interested in reconstructing a network from partial observations or predicting future interactions. LinkPred offers two ways to solve such problems, computing the scores of all non-existing links and computing top $k$ edges, which may be more efficient for large networks. This section demonstrates how to perform both tasks.

The following code excerpt shows how to compute and print the scores of all non-existing links in a network using SBM. The observed network is passed as an argument to the constructor of the algorithm, which is then initialized by calling the method init. The learning process, if any, is triggered by a call to the method learn. The simplest way to

obtain the score of a non-existing link is to call the method score, though other methods of the predictor interface may result in better performance.

```cpp
#include <linkpred.hpp>
#include <iostream>
using namespace LinkPred;
int main() {
  // Read the network from file
  auto net = UNetwork<>::read("Zakarays_Karate_Club.edges");
  // Create an SBM predictor (777 is a seed)
  USBMPredictor<> predictor(net, 777);
  // Initialize predictor
  predictor.init();
  // Train predictor
  predictor.learn();
  // Print scores
  std::cout << "#Start\tEnd\tScore\n";
  for (auto it=net->nonEdgesBegin();it!=net->nonEdgesEnd();++it){
    auto i = net->getLabel(net->start(*it));
    auto j = net->getLabel(net->end(*it));
    double sc = predictor.score(*it);
    std::cout << i << "\t" << j << "\t" << sc << std::endl;
  }
  return 0;
}
```

The first few lines of the output of this program are as follows:

```
#Start  End  Score
1   31  0.208413
1   10  0.248398
1   28  0.229615
1   29  0.246439
1   33  0.316544
1   17  0.685567
1   34  0.315658
1   15  0.17834
1   16  0.178189
...
```

We can also use an embedding-classifier predictor. In the following code, the graph is embedded using node2vec and logistic regression is used to predict scores.

```cpp
#include <linkpred.hpp>
#include <iostream>
using namespace LinkPred;
int main() {
  // Read the network from file
  auto net = UNetwork<>::read("Zakarays_Karate_Club.edges");
  // Create a node2vec encoder (777 is a seed)
  auto encoder = std::make_shared<Node2Vec<>>(net, 777);
  // Create a logistic regresser (0.001 is the regularization coefficient, and 888
      is a seed)
  auto classifier = std::make_shared<LogisticRegresser<>>(0.001, 888);
  // Create an encoder-classifier predictor (999 is a seed)
  UECLPredictor<> predictor(net, encoder, classifier, 999);
  // Initialize predictor
  predictor.init();
  // Train predictor
  predictor.learn();
  // Print scores
  std::cout << "#Start\tEnd\tScore\n";
  for (auto it=net->nonEdgesBegin();it!=net->nonEdgesEnd();++it){
```

```
      auto i = net->getLabel(net->start(*it));
      auto j = net->getLabel(net->end(*it));
      double sc = predictor.score(*it);
      std::cout << i << "\t" << j << "\t" << sc << std::endl;
   }
   return 0;
}
```

The following is partial output of this program:

```
#Start  End  Score
1   31   0.415466
1   10   0.506023
1   28   0.240406
1   29   0.417463
1   33   0.336364
1   17   0.390554
1   34   0.741211
1   15   0.203656
1   16   0.256206
...
```

Instead of computing the scores of all non-existing links, it is possible to extract the top $k$ ranked edges only. Besides convenience, this approach may be the only viable option for very large networks due to memory considerations. Furthermore, for many prediction algorithms, particularly topological similarity methods, finding the top $k$ edges is much faster than computing the scores of all non-existing links. The following code shows how to find the top $k$ edges using Resource Allocation index.

```
#include <linkpred.hpp>
#include <iostream>
using namespace LinkPred;
int main() {
   int k = 10;
   // Read the network from file
   auto net = UNetwork<>::read("Zakarays_Karate_Club.edges");
   // Create a RAL predictor
   URALPredictor<> predictor(net);
   // Initialize predictor
   predictor.init();
   // Train predictor
   predictor.learn();
   // Allocate memory
   std::vector<typename UNetwork<>::Edge> edges(k);
   std::vector<double> scores(k);
   // Find top k edges
   k = predictor.top(k, edges.begin(), scores.begin());
   // Print edges and scores
   std::cout << "#Start\tEnd\tScore\n";
   for (int l = 0; l < k; l++) {
      auto i = net->getLabel(net->start(edges[l]));
      auto j = net->getLabel(net->end(edges[l]));
      std::cout << i << "\t" << j << "\t" << scores[l] <<std::endl;
   }
   return 0;
}
```

This is the output of this program:

```
#Start  End  Score
1  17  0.5
1  34  0.9
```

```
2   34  0.783333
3   32  0.479167
3   34  1.56667
5   6   0.645833
7   11  0.645833
32  24  0.47549
28  26  0.533333
24  25  0.583333
```

### Implementing a new link prediction algorithm

The first step in implementing a new link prediction algorithm is to inherit from `ULPredictor` and implement the necessary methods. For a minimal implementation, the three methods `init`, `learn` and `score` must at least be defined. To achieve better performance one may want to redefine the three other methods (`top`, `predict` and `predictNeg`).

Suppose one wants to create a very simple link prediction algorithm that assigns as score to $(i,j)$ the score $\kappa_i + \kappa_j$, the sum of the degrees of the two nodes. In a file named `sdpredictor.hpp`, write the following code:

```cpp
#ifndef SDPREDICTOR_HPP_
#define SDPREDICTOR_HPP_
#include <linkpred.hpp>
class SDPredictor: public LinkPred::ULPredictor<> {
  using LinkPred::ULPredictor<>::net;
  using LinkPred::ULPredictor<>::name;
public:
  using Edge = typename LinkPred::ULPredictor<>::Edge;
  SDPredictor(std::shared_ptr<LinkPred::UNetwork<> const> net) : LinkPred::
      ULPredictor<>(net) {
    name = "SD";
  }
  virtual void init();
  virtual void learn();
  virtual double score(Edge const & e);
  virtual ~SDPredictor() = default;
};
#endif
```

In a file named `sdpredictor.cpp` write the implementation of the inherited methods (note that this predictor does not require initialization or learning):

```cpp
#include "sdpredictor.hpp"
// No init required
void SDPredictor::init() {}
// No training required
void SDPredictor::learn() {}
// Here, we compute the score
double SDPredictor::score(Edge const & e) {
  auto i = net->start(e);
  auto j = net->end(e);
  // Return the sum of degrees
  return net->getDeg(i) + net->getDeg(j);
}
```

This predictor is now ready to be used with LinkPred classes and methods including performance evaluating routines. For instance, it is possible to write a code that extracts the edges with the top scores as follows:

```cpp
#include "sdpredictor.hpp"
#include <iostream>
using namespace LinkPred;
int main() {
  std::size_t k = 10;
  // Read network from file
  auto net = UNetwork<>::read("Zakarays_Karate_Club.edges");
  // Create predictor
  SDPredictor predictor(net);
  // Initilize predictor
  predictor.init();
  // Train predictor
  predictor.learn();
  // Allocate memory
  std::vector<typename UNetwork<>::Edge> edges(k);
  std::vector<double> scores(k);
  // Find top k edges
  k = predictor.top(k, edges.begin(), scores.begin());
  // Print edges and scores
  std::cout << "#Start\tEnd\tScore\n";
  for (int l = 0; l < k; l++) {
    auto i = net->getLabel(net->start(edges[l]));
    auto j = net->getLabel(net->end(edges[l]));
    std::cout << i << "\t" << j << "\t" << scores[l] <<std::endl;
  }
  return 0;
}
```

Upon compiling and executing this code, the output will be as follows (for compilation instructions, the reader is invited to consult the library user guide):

```
#Start  End  Score
1   33   28
1   34   33
1   24   21
2   33   21
2   34   26
3   34   27
4   34   23
6   34   21
7   34   21
8   34   21
```

New link prediction algorithms can also be easily integrated into the library source code, as explained in detail in the library user guide.

## Performance evaluation

Another use case scenario is evaluating and comparing the performance of link prediction algorithms. LinkPred offers several ways to achieve this, offering various degrees of control on the evaluation process. One such method is shown in the code sample below. Here, the user defines a factory class used to instantiate the prediction algorithms and performance measures. The parameters of the experiment, including the ratio of removed edges and the number of test runs, are passed through an object of type `PerfeEvalExpDescp`. The evaluation is finally conducted by passing the factory and parameter objects to an object of type `PerfEvalExp` then calling the method run.

```cpp
#include <linkpred.hpp>
using namespace LinkPred;
// This class is used to create predictors and performance measures
```

```cpp
class Factory: public PEFactory<> {
public:
  // Create predictors
  virtual std::vector<std::shared_ptr<ULPredictor<>>> getPredictors(std::
      shared_ptr<UNetwork<> const> obsNet) {
    std::vector<std::shared_ptr<ULPredictor<>>> prs;
    // Add predictors
    prs.push_back(std::make_shared<URALPredictor<>>(obsNet));
    prs.push_back(std::make_shared<UKABPredictor<>>(obsNet));
    return prs;
  }
  // Create performance measures
  virtual std::vector<std::shared_ptr<PerfMeasure<>>> getPerfMeasures(TestData<>
      const & testData) {
    std::vector<std::shared_ptr<PerfMeasure<>>> pms;
    // Add top-precision
    pms.push_back(std::make_shared<TPR<>>(testData.getNbPos()));
    // Add AUCROC
    pms.push_back(std::make_shared<ROC<>>());
    return pms;
  }
  virtual ~Factory() = default;
};
int main() {
  // Read reference network from file
  auto refNet = UNetwork<>::read("Zakarays_Karate_Club.edges");
  // Description of the experiment
  PerfeEvalExpDescp<> ped;
  ped.refNet = refNet;
  ped.nbTestRuns = 10;
  ped.seed = 777;
  // Create the factory object
  auto factory = std::make_shared<Factory>();
  // Create the experiment object
  PerfEvalExp<> exp(ped, factory);
  // Run the experiment
  exp.run();
  return 0;
}
```

The output results for the first few iterations is a s follows:

```
#ratio  ROCKAB  ROCRAL  TPRKAB  TPRRAL
0.10    0.8615  0.8028  0.1250  0.1250
0.10    0.7943  0.7823  0.1250  0.1667
0.10    0.6945  0.6712  0.0000  0.0000
0.10    0.6417  0.6219  0.2500  0.1250
0.10    0.5817  0.5487  0.0000  0.0000
0.10    0.8527  0.8386  0.3750  0.3438
0.10    0.5705  0.5167  0.0000  0.0000
0.10    0.8834  0.8359  0.1250  0.1250
0.10    0.8962  0.8617  0.2500  0.1250
0.10    0.7650  0.7433  0.2500  0.2500
```

More use case examples can be found in the library documentation. These include using
other link prediction algorithms, computing the scores of a specific set of edges, and other
methods for computing the performance of one or several link prediction algorithms.

## EXPERIMENTAL RESULTS

In addition to providing an easy interface to use, create and evaluate link prediction
algorithms, LinkPred is designed to handle very large networks, which is a quality that is

essential for most practical applications. To demonstrate the performance of LinkPred, its time performance is compared to that of the R package *linkprediction* and the Python packages linkpred, NetworkX and scikit-network. To conduct a fair and meaningful comparison, two issues are to be resolved. First, these packages do not implement the same set of algorithms, and only a limited number of topological similarity methods are implemented by all five libraries. Accordingly, the Resource Allocation index is chosen as the comparison task, since it is implemented by all five packages and exhibits the same network data access patterns as most local methods. The second issue that needs to be addressed is that the libraries under consideration offer programming interfaces with different semantics. For instance, scikit-network computes the score for edges given as input, whereas the R package *linkprediction* and Python packages Linkpred and NetworkX do not require input and instead return the scores of non-existing links. Furthermore, the Python package linkpred returns the scores of only candidate edges that have a non-zero score. To level the field, the comparison shall consist in computing the scores of all non-existing links, even those with zero scores. All networks used in this experiment are connected due to the restriction imposed by the package *linkprediction*. A description of these networks is given in Table 4 of the appendix. For the sake of fairness, parallelism is disabled in LinkPred, and all experiments are conducted on a single core of an Intel Core i7-4940MX CPU with 32GB of memory. The time reported in Table 2 is the average execution time over ten runs, excluding the time required to read the network from file. The time for LinkPred is reported for C++ code and the Java and Python bindings. The results show that LinkPred is typically one to two orders of magnitudes faster than the other packages. This, of course, can in part be explained by the interpreted nature of Python and R, but it also highlights the fact that link prediction is a computationally intensive task that is best handled by high-performance software that uses efficient data structures and algorithms. As shown in the table, the Java binding of LinkPred introduces a small overhead compared to its Python binding due to more complex data marshaling in the latter. Nevertheless, the Python binding is significantly faster than the Python packages and, except for a couple of networks, is also faster than *linkprediction*.

Table 3 shows the time taken by LinkPred to complete different link prediction tasks on various hardware architectures. It shows that the library can handle very large networks in relatively small amounts of time, even when the available computational resources are limited.

## CONCLUSION AND FUTURE WORK

LinkPred is a distributed and parallel library for link prediction in complex networks. It contains the implementation of the most important link prediction algorithms found in the literature. The library is designed not only to achieve high performance but also to be easy-to-use and extensible. The experiments show that the library can handle very large networks with up to millions of nodes and edges and is one to two orders of magnitude faster than existing Python and R packages. LinkPrted components interact through clearly defined and easy interfaces, allowing users to plug their own components into the library by

**Table 2  Time (in seconds) required to compute the score of all non-existing links using Resource Allocation index on a single core.**

| Network | LinkPred (C++) | LinkPred (Java) | LinkPred (Python) | Python package NetworkX | R package *linkprediction* | Python package linkpred | Python package scikit-network |
|---|---|---|---|---|---|---|---|
| Political Blogs | 0.02 | 0.03 | 0.14 | 1.83 | 3.70 | 0.68 | 3.13 |
| Diseasome | 0.04 | 0.16 | 0.86 | 6.33 | 2.53 | 1.26 | 14.98 |
| Email | 0.05 | 0.12 | 0.56 | 7.78 | 6.88 | 1.60 | 9.63 |
| Web Edu | 0.14 | 0.72 | 4.16 | 36.92 | 8.67 | 5.31 | 68.71 |
| Java | 0.08 | 0.23 | 1.04 | 17.08 | 55.54 | 8.95 | 17.82 |
| Power | 0.36 | 1.83 | 11.05 | 80.55 | 3.80 | 11.16 | 183.71 |
| Erdos 02 | 0.76 | 3.62 | 21.71 | 179.75 | 44.15 | 30.42 | 358.37 |
| World Air | 0.31 | 1.10 | 5.79 | 81.06 | 55.06 | 11.71 | 97.91 |
| Oregon | 2.32 | 9.62 | 56.96 | 525.76 | 573.47 | 157.60 | 936.84 |
| PGP | 2.42 | 9.12 | 51.31 | 603.75 | 35.74 | 57.32 | 862.56 |
| Spam | 0.99 | 2.33 | 10.33 | 318.16 | 199.83 | 42.80 | 171.68 |
| Indochina 2004 | 2.48 | 10.04 | 59.16 | 1,086.26 | 91.95 | 74.61 | 1,003.82 |

**Table 3  Time achieved by LinkPred on different prediction tasks.**  Column $n$ contains the number of nodes in the network, whereas $m$ shows the number of edges.

| Network | $n$ | $m$ | Task | Hardware | Time (sec.) |
|---|---|---|---|---|---|
| Brightkite | 58,228 | 214,078 | Compute ROC using 10% removed edges for ADA. | 1 node, 6 cores (Core i7-8750H) | 32.92 |
| Yahoo IM | 100,001 | 587,964 | Find the top $10^4$ edges using RAL. | 1 node, 1 core (Core i7-8750H) | 6.70 |
| Twitter | 404,719 | 713,319 | Find the top $10^5$ edges using RAL. | 1 node, 1 core (Core i7-8750H) | 16.93 |
| Youtube | 1,134,890 | 2,987,624 | Find the top $10^5$ edges using CNE. | 1 node, 6 cores (Core i7-8750H) | 79,41 |
| CA Roads | 1,965,206 | 2,766,607 | Find the top $10^5$ edges using CNE. | 1 node, 6 cores (Core i7-8750H) | 7.08 |
| Wiki Talks | 2,394,385 | 4,659,565 | Find the top $10^5$ edges using CNE. | 1 node, 6 cores (Core i7-8750H) | 470.04 |
| Internet | 124,651 | 193,620 | Compute top-precision using 10% removed edges for eight algorithms. | 8 nodes, 16 cores in each node (Xeon E5-2650) | 3.73 |
| Amazon | 334,863 | 925,872 | Compute top-precision using 10% removed edges for eight algorithms. | 8 nodes, 16 cores in each node (Xeon E5-2650) | 24.17 |

implementing these interfaces. In particular, users can integrate their own link prediction algorithms and performance measures seamlessly into the library. This makes LinkPred an ideal tool for practitioners as well as researchers in link prediction.

The library can be improved and extended in several ways, such as adding R and Octave/Matlab bindings. Another possibility for improvement is implementing further

**Table 4  Description of the networks used in the experimental analysis.** Columns *n* and *m* represent the number of nodes and edges in the network, respectively.

| Network | Description | n | m |
| --- | --- | --- | --- |
| Amazon (*Yang & Leskovec, 2015*) | Amazon product co-purchasing network. An edge indicates that two products have been co-purchased. Data available at https://snap.stanford.edu/data/com-Amazon.html. | 334,863 | 925,872 |
| Brightkite (*Cho, Myers & Leskovec, 2011*) | Friendship network on the social platform Brightkite. The data is available at https://snap.stanford.edu/data/loc-Brightkite.html. | 58,228 | 214,078 7 |
| CA Roads (*Leskovec et al., 2009*) | California road network. Data available at https://snap.stanford.edu/data/roadNet-CA.html. | 1,965,206 | 2,766,607 |
| Diseasome (*Goh et al., 2007*) | A network of genes' disorders and disease linked by known disorder–gene associations. The data is available at http://gephi.org/datasets/diseasome.gexf.zip. | 1,419 | 2,738 |
| Email (*Guimerà et al., 2003*) | The symmetrized network of email communication at the University Rovira i Virgili (Tarragona, Spain). The nodes represent users, and edges indicate an email communication took place between the two uses. The dataset is available at http://deim.urv.cat/~alexandre.arenas/data/welcome.htm. | 1,133 | 5,451 |
| Erdos 02 | The 2002 version of Erdös' co-authorship network. The network is available at http://vlado.fmf.uni-lj.si/pub/networks/data/Erdos/Erdos02.net. | 6,927 | 11,850 |
| Indochina 2004 (*Boldi & Vigna, 2004*; *Boldi et al., 2011*) | A WWW network available at http://networkrepository.com/web_indochina_2004.php. | 11,358 | 47,606 |
| Internet (*Batagelj & Mrvar, 2006*) | Network of Internet routers. The network is available at https://sparse.tamu.edu/Pajek/internet. | 124,651 | 193,620 |
| Java | The symmetrized version of a network where nodes represent Java classes and edges represent compile-time dependencies between two classes. The dataset can be found at http://vlado.fmf.uni-lj.si/pub/networks/data/GD/GD.htm. | 1,538 | 7,817 |
| Oregon (*Leskovec, Kleinberg & Faloutsos, 2005*) | Autonomous Systems (AS) peering network inferred from Oregon route-views on May 26, 2001.The data is available at https://snap.stanford.edu/data/oregon1_010526.txt.gz. | 11,174 | 23,409 |
| PGP (*Boguñá et al., 2004*) | A social network of users using Pretty Good Privacy (PGP) algorithm. The network is available at http://deim.urv.cat/~alexandre.arenas/data/welcome.htm. | 10,680 | 24,316 |
| Political Blogs (*Adamic & Glance, 2005*) | A network of hyperlinks among political web blogs. The data is available at http://networkrepository.com/web-polblogs.php. | 643 | 2,280 |
| Power (*Watts & Strogatz, 1998*) | The Western States Power Grid of the United States. Data available at http://www-personal.umich.edu/~mejn/netdata/. | 4,941 | 6,594 |
| Spam (*Castillo, Chellapilla & Denoyer, 2008*) | A WWW network available at http://networkrepository.com/web-spam.php. | 4,767 | 37,375 |
| Twitter (*Gleich & Rossi, 2014*) | A Twitter network of follow relationship. Data available at http://networkrepository.com/soc-twitter-follows.php. | 404,719 | 713,319 |

**Table 4** (*continued*)

| Network | Description | n | m |
|---|---|---|---|
| Web Edu (*Gleich, Zhukov & Berkhin, 2004*) | A WWW network available at http://networkrepository.com/web-edu.php. | 3,031 | 6,474 |
| Wiki Talks (*Leskovec, Huttenlocher & Kleinberg, 2010b*; *Leskovec, Huttenlocher & Kleinberg, 2010a*) | A symmetrized version of the Wikipedia talk network. A node represents a user, and an edge indicates that one user edited the talk age of another user. Data available at https://snap.stanford.edu/data/wiki-Talk.html. | 2,394,385 | 4,659,565 |
| World Transport (*Guimerà et al., 2005*) | A worldwide airport network. Nodes represent cities, and edges indicate a flight connecting two cities. The data is available at http://seeslab.info/media/filer_public/63/97/63979ddc-a625-42f9-9d3d-8fdb4d6ce0b0/airports.zip. | 3,618 | 14,142 |
| Yahoo IM (*Yahoo! Webscope, 2008*) | Network of sample Yahoo! Messenger communication events. The data is available at https://webscope.sandbox.yahoo.com/catalog.php?datatype=g. | 100,001 | 587,964 |
| Youtube (*Yang et al., 2015*) | A Youtube friendship network. Data available at https://snap.stanford.edu/data/com-Youtube.html. | 1,134,890 | 2,987,624 |
| Zakary's Karate Club (*Zachary, 1977*) | A friendship network among members of a karate club at an American university. The data was collected in the 1970s by Wayne Zachary and is available at http://konect.cc/networks/ucidata-zachary | 34 | 78 |

graph embedding algorithms, particularly those based on deep neural networks. Also important is handling dynamic (time-evolving) networks. Finally, sampling-based methods such as SBM and FBM, although producing good results, are only usable with small networks because they are computationally intensive. Distributed implementations of these algorithms will allow using them in practical situations on large networks.

## ACKNOWLEDGEMENTS

This research work is supported by the Research Center, CCIS, King Saud University, Riyadh, Saudi Arabia. Part of the computational experiments reported in this work were conducted on the SANAM supercomputer at King Abdulaziz City for Science and Technology (https://www.kacst.edu.sa/).

## APPENDIX: DATA

Table 4 contains the description of all networks used in the experimental evaluation section.

### Funding

This research work is supported by the Research Center, CCIS, King Saud University, Riyadh, Saudi Arabia. The funders had no role in study design, data collection and analysis, decision to publish, or preparation of the manuscript.

## Grant Disclosures

The following grant information was disclosed by the author:
Research Center, CCIS, King Saud University, Riyadh, Saudi Arabia.

## Competing Interests

The authors declare there are no competing interests.

## Author Contributions

- Said Kerrache conceived and designed the experiments, performed the experiments, analyzed the data, performed the computation work, prepared figures and/or tables, authored or reviewed drafts of the paper, and approved the final draft.

## Data Availability

  Data and code are available in the Supplementary Files.

## Supplemental Information

Supplemental information for this article can be found online at http://dx.doi.org/10.7717/peerj-cs.521#supplemental-information.

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
