# Peer review of "LinkPred: a high performance library for link prediction in complex networks"

_PeerJ Computer Science, doi:10.7717/peerj-cs.521_

## Round 0.1 · original submission · Major Revisions

Although your library seems useful and with potential, there are some weaknesses that should be fixed before publishing, especially documentation..

·

Basic reporting

This paper proposed a library with ease of use interfaces for link prediction. It reported and reviewed briefly the current existing related works' limitations that lead to LinkPred development. It gave a detail explanation on how the LinkPred was produced and deployed. The report clearly explained how the development of the LinkPred and showed the result of its time execution performance on eight different networks as depicted in Table 1. However some questions arise - From Figure 1, how 'user-defined' can be deployed together with the LinkPred library? How different is between 'user defined' in link prediction algorithm with performance measures? How the work prove ease of use, extensibility, and efficiency as claimed in line 37? Why Djikstra's algo is chosen? ULPredictor must be explained further as what method has been used to predict? why it has been chosen? How accurate the prediction result is?

Experimental design

The experimental design started by showing the architectural design in general explanation followed by the development of the library. Research questions are not provided with that may lead to many questions on the development of the library. However, the elaboration of the development is very useful for researchers to understand how the library works. A section on how users apply the library in their scenario will be very beneficial.

Validity of the findings

Thorough experiments to prove the library's ease of use, extensibility and efficiency is needed to improve the paper further, despite its clear explanation on the library development. How the LinkPred library can be used in different programming languages that affect effectiveness? It is believed the conclusion will be well stated when most of these questions answered.

Additional comments

The work proposed a library for link prediction which is very beneficial and relevant in the current network situation. The library will benefit practitioners and researchers to fasten up their work process as the network is scaled every day. However, the proposed library needs more work for improvements as questions arise especially in term of its reliability.

Reviewer 2 ·

Basic reporting

References to key libraries doing link prediction is missing including SNAP (http://snap.stanford.edu/snap/description.html) and GEM (https://github.com/palash1992/GEM).

The authors have not considered graph embedding/representation learning based approaches for link prediction which have been very popular recently.

The documentation in the library is very limited.

Experimental design

The approaches included in the library are not comprehensive missing several state-of-the-art link prediction methods including those based on graph representation learning which are included in libraries such as SNAP an GEM.

Validity of the findings

The methods implemented have been duly cited and well defined.

---

## Round 0.2 · accepted · Accept

Both reviewers consider that raised issues have successfully addressed. I am pleased to inform you that your paper can be accepted for publication.

·

Basic reporting

No comment.

Experimental design

No comment

Validity of the findings

No comment

Additional comments

The latest version of the paper has very significant improvement which clearly answered the previous questions with good explanation by additional data and experiments. Important and relevant works of literature are included where necessary in each section. The research methodology is clear. The library construction is clear and justified. The issue of the library's reliability has been answered well.

Reviewer 2 ·

Basic reporting

The authors have addressed all my concerns.

Experimental design

The authors have addressed all my concerns.

Validity of the findings

The authors have addressed all my concerns.

Additional comments

Thanks for addressing all the concerns. The manuscript and code repository look good now.